# Pragmatic De-Noising of Electroglottographic Signals

**DOI:** 10.3390/bioengineering11050479

**Published:** 2024-05-11

**Authors:** Sten Ternström

**Affiliations:** Division of Speech, Music and Hearing, School of Electrical Engineering and Computer Science, KTH Royal Institute of Technology, 100 44 Stockholm, Sweden; stern@kth.se; Tel.: +46-8-790-7855

**Keywords:** electroglottography, de-noising, contact quotient, peak dEGG, spectral thresholding, notch filtering

## Abstract

In voice analysis, the electroglottographic (EGG) signal has long been recognized as a useful complement to the acoustic signal, but only when the vocal folds are actually contacting, such that this signal has an appreciable amplitude. However, phonation can also occur without the vocal folds contacting, as in breathy voice, in which case the EGG amplitude is low, but not zero. It is of great interest to identify the transition from non-contacting to contacting, because this will substantially change the nature of the vocal fold oscillations; however, that transition is not in itself audible. The magnitude of the cycle-normalized peak derivative of the EGG signal is a convenient indicator of vocal fold contacting, but no current EGG hardware has a sufficient signal-to-noise ratio of the derivative. We show how the textbook techniques of spectral thresholding and static notch filtering are straightforward to implement, can run in real time, and can mitigate several noise problems in EGG hardware. This can be useful to researchers in vocology.

## 1. Introduction

In voice analysis, the electroglottographic (EGG) signal has long been recognized [1] as a useful electromechanical complement to the acoustic signal. For the purpose, principle, and interpretation of EGG, the reader is referred to existing overviews [2,3]. 

It is commonly held that the EGG signal is useful mainly when the vocal folds (VFs) are actually contacting, such that the variations in vocal fold contact area (VFCA) have an appreciable amplitude. However, phonation can also occur without VF contacting, as in soft or breathy voice. Identifying the transition from non-contacting to contacting is of great theoretical and clinical interest, because the onset of contacting substantially changes the nature of the vocal fold oscillations, with consequences for numerous metrics of the voice. Interestingly, this transition between contacting and non-contacting is not audible as such. In a separate article we intend to describe how EGG metrics can indicate contacting, how VF contacting affects other voice metrics, and how non-contacting VF oscillations dominate the considerable parts of the voice range that are low in sound pressure level (SPL). The present article is a Technical Note on how some conventional de-noising techniques can clarify the transition to VF contacting that otherwise tends to be obscured by noise.

In soft or breathy phonation, the vocal folds oscillate without contacting, and the EGG amplitude is very low. It is not quite zero—because there is still a tiny variation in VFCA at one or both ends of the vocal folds. This ‘micro-variation’ tends to be very nearly sinusoidal. A convenient indicator of VF contacting is the cycle-normalized peak *Q*_Δ_ of the time derivative of the EGG signal (dEGG) [4], which assumes a minimum value of 1 for the sinusoidal waveform of no contact and abruptly increases whenever contact occurs. However, this holds only if the signal is not contaminated by noise, as will be explained below. 

Sources of noise in EGG signals have been discussed by a few authors, notably Rothenberg [5]. Titze [3] considered the mechanism of electrical transduction through the body surrounding the larynx, and its general influence on the signal-to-noise ratio (SNR) of the EGG signal. The peak dEGG and its interpretations have been discussed by several authors [6,7], but to our knowledge, the influence of noise on the derivative of the EGG signal has not received much attention. The few existing manufacturers of EGG hardware have hitherto focused entirely on the EGG waveform during contacting and have striven to achieve an SNR that is adequate for representing high-amplitude signals. This is quite a challenge in itself. In practice, many models on the market have a poor SNR even for signals with distinct contacting, which hampers the precise estimation of waveform characteristics such as the cycle period, contact quotient, and peak derivative. This was recognized by Herbst and Dunn [8], who compared the performance of several fundamental frequency (*f*_o_) estimation algorithms in the presence of typical kinds of EGG noise. 

If we want reliable detection of a low-amplitude modulation of the VFCA, there is currently no commercial hardware that we are aware of that affords a sufficiently high SNR of the signal’s *derivative*, which is necessarily much noisier than the signal itself. We therefore seek to suppress some common kinds of noise in EGG signals post hoc. While there is a body of literature on the de-noising of biophysical signals, e.g., [9], there appears to be only a few existing studies specifically on de-noising the EGG signal. Bouzid and Ellouze [10] applied the wavelet transform with multiscale products and reported good results for several EGG features, even with noisy signals. Colominas et al. [11] presented an ambitious approach based on empirical mode decomposition of biophysical signals, including applications to the EGG signal. Deshpande et al. [12] proposed a variational mode decomposition approach with five stages for detecting glottal instants specifically from noisy EGGs, and found it to perform better than several other methods. Still, all of these methods require considerable expertise and manual adjustment to implement and operate. 

The present article is instead a Technical Note on how conventional post hoc signal-processing methods from the audio domain can be applied to pragmatically reduce the impact of some noise sources that may be present in EGG signals. Here, ‘pragmatic’ means that the methods can be applied as part of an existing analysis chain without requiring much extra time or expertise from the operator. The primary aim here is not to propose an optimal solution for removing all noise from an EGG signal, but mainly to enable the detection of VF contacting, from the normalized peak dEGG being greater than 1. First, several types of noise are reviewed, then some possible remedies are proposed, and finally, a practical implementation is described.

## 2. Types of Noises in EGG Signals

EGG signals are very often contaminated by noise from several sources. These include (1) low-frequency drift due to relatively slow conductance changes in the body of the informant, (2) electrical hum from the device’s own power supply or from the general electromagnetic environment of the experimental setup, (3) broadband system noise of analog or digital origin, and (4) rogue static high frequencies in the 2–20 kHz range. This inventory of noise types is quite similar to that for other biophysical signals, such as the EMG [13]. When preparing for data collection with the EGG signal, it is therefore essential to first assess the signal quality by *listening* to the EGG signal and also to inspect it with full-range, narrow-band spectra and spectrograms. 

### 2.1. Low-Frequency Drift

While the noise types to be described below all stem from the equipment, even the EGG signal itself contains unwanted components. It is the AC component of the EGG signal that we are interested in, which represents the change in VFCA during phonation. This AC component is however much smaller in amplitude than the total EGG signal amplitude (typically 1–10%). EGG signals tend to contain a large amount of irrelevant low-frequency fluctuations at <20 Hz, ‘near-DC’ (Figure 1), which are due to conductance variations from muscle activity and sometimes even from the pulsating blood flow in the neck. These near-DC components can make automatic segmentation into cycles [4] less accurate, and they can easily offset the signal by so much that clipping occurs, thereby corrupting or even obliterating the AC component. Therefore, most EGG devices include an analog high-pass filter at 2–20 Hz that attenuates the near-DC parts of the signal. This filtering inevitably introduces some distortion of the EGG pulse shape, especially when *f*_o_ is low. Some systems can perform post-compensation for such distortion [14]. 

### 2.2. Broadband Noise

Any analog electronic equipment will have some inherent background thermal noise, while digital systems necessarily have quantization noise. Both these noise types are usually evenly spread across the spectrum. However, the dynamic range of the AC component of the EGG signal is typically no larger than 40 dB, even in the best hardware. This means that the analog noise of the EGG device will usually be much stronger than the digital quantization noise introduced by a standard 16-bit A/D converter. (For the acoustic signal, the dynamic range of trained human voices can approach or exceed 90 dB, which requires 24-bit conversion and very-low-noise preamplifiers, in order to stay well above the quantization noise floor.)

Some EGG devices have rechargeable lead–acid battery cells, which may start to generate a crackling noise if they are overcharged. Once it has set in, this type of noise does not go away. The cells will need to be replaced, as instructed by the manufacturer. 

### 2.3. Spurious High Frequencies

More often than not, a narrow-band long-time average spectrum (LTAS) of the EGG signal will reveal one or more static frequencies at 2 kHz or more, superimposed on the signal. An example is shown in Figure 2 of a quiescent EGG channel when there is no phonation. Such ‘side tones’ can result from sampling artifacts, switching power supplies, digital/analog crosstalk within the device, or interference with other equipment that, like the EGG, modulates signals in the MHz range, such as inductive bands for respiration measurements. High-frequency spikes in the frequency domain are particularly disruptive to the estimation of *Q*_Δ_ in the time domain. If the frequency of a side tone is static, the tone can usually be suppressed using a narrow notch filter, as described below.

For convenience, current EGG devices often have a low-cost stereo USB audio interface built in, the EGG signal being paired with the microphone signal. This is not without its drawbacks, though, because (a) the dynamic range of the accompanying microphone preamplifier is rarely large enough for the voice; and (b) such devices are prone to digital/analog crosstalk, which is a source of side tones. In the current market, this author prefers analog-only EGG devices and digitizing their signals with a separate high-quality digital audio interface. 

### 2.4. Electrical Hum

Power line interference, or mains-induced hum, is a stationary tone at 50 or 60 Hz, with harmonics at integer multiples of that fundamental frequency, sometimes extending as high as 500 Hz. This is in the same range as that of the EGG signals of human phonation, which can be problematic if the hum is strong. An example is shown in Figure 3. 

## 3. Signal Pre-Processing Techniques

### 3.1. High-Pass Filtering

EGG devices usually have some kind of analog high-pass filter built in, to attenuate the large unwanted near-DC component. Still, the remaining low-frequency content can complicate the computation of the contact quotient. In the digital domain, recursive filters (infinite impulse response, IIR) compute quickly but tend to introduce phase distortion even above the cutoff frequency. Instead, it is possible to apply a very steep digital linear-phase transverse filter (finite impulse response, FIR) to eliminate the near-DC component almost entirely, which simplifies further processing. The disadvantage of such steep filtering is that if the *f*_o_ descends below the cutoff frequency, there will be some distortion of the EGG waveform. Also, such a filter is necessarily long and thus introduces a delay that may be noticeable. To achieve low latency for real-time feedback applications, this means that its cutoff frequency cannot be lower than about 100 Hz if the sampling rate is 44,100 Hz. An example is shown in the left-hand part of Figure 2. This filter was designed as a 1024-point FIR high-pass filter with a cutoff frequency of 100 Hz and a stopband attenuation below 20 Hz of −60 dB.

### 3.2. Notch Filtering

In Figure 2, the blue curve shows an LTAS of a quiescent EGG signal with two static side tones at 2850 at 6080 Hz, and also a narrow noise band centered on 3520 Hz. Taking the derivative of a signal is equivalent to tilting its spectrum up by +20 dB per decade in frequency, so these unwanted signal components will cause problems when estimating the aforementioned *Q*_Δ_ metric. A so-called ‘notch’ filter can be used to attenuate them selectively. A notch filter results from setting a parametric equalizer filter to negative gain and a narrow bandwidth. It can be implemented as a single second-order section. An example of using three such filters is given by the orange curve in Figure 2. The frequencies, bandwidths, and gains were manually adjusted by inspection of the narrow-band spectrum, so as to suppress the offending tones to below the general noise floor. This will work only if the side tones are very stable in frequency over the duration of the recording. The EGG waveform itself normally contains very little energy at these frequencies, so the impact of notch filtering here on any EGG shape parameters is negligible.

### 3.3. Spectral Thresholding

If broadband noise has equal power at all frequencies, it is said to be ‘white’. Such noise can be effectively reduced using the textbook technique of spectral thresholding (also known as spectral gating), as follows. The signal to be de-noised is first transformed into the frequency domain, with an analysis bandwidth that is several times smaller than *f*_o_, say 20 Hz (Figure 4a). A level threshold is then applied to the log power spectrum such that only the important EGG harmonics exceed this threshold. In the frequency bins whose levels are below this threshold, the magnitudes are attenuated downwards with a ratio of 4:1 in dB relative to the threshold (dynamics ‘expansion’), while the phases are left unchanged. Expansion is preferable to zeroing in order to avoid transients when individual frequency components cross the threshold. The result is that any noise whose level is below the threshold, i.e., between the harmonics, is attenuated; while those harmonics whose levels are above the threshold are not attenuated (Figure 4b). 

The resulting spectrum is then inverse transformed back to the time domain. This can all be conducted in real time with a total buffering delay that is in the order of 80 ms. The effect on noisy EGG waveforms is to make them smooth without distorting the EGG pulse shape. This makes it possible to take the derivative as sample-to-sample differences, where the peak value in a cycle is not corrupted by noise.

The difference between low-pass filtering and spectral thresholding is further illustrated in Figure 5 with a synthesized sawtooth sweep to which white noise is added. It can be seen that a low-pass filter at 5 kHz removes the highest frequencies only, while the thresholding suppresses noise in between the harmonics. In the latter case, the weaker high harmonics are still lost, but they contribute very little to the shape of the EGG.

Better noise rejection can be obtained by narrowing the analysis bandwidth, but this also reduces the temporal resolution. Changes in *f*_o_ within a long analysis time window may cause the spectrum envelope between the harmonics to rise above the threshold, reducing the suppression of noise. With a constant analysis bandwidth, the noise rejection improves when *f*_o_ increases since there is more vacant space between the harmonics. 

If the noise is not white, but unequally distributed across the spectrum, it becomes more complicated to suppress it in this way. Custom solutions, such as a frequency-dependent threshold, then have to be devised for the particular noise. 

### 3.4. Hum Abatement

The harmonics of mains-induced hum are typically close in frequency to those of the EGG signal, so it is usually not possible to filter or otherwise post-process them out without distorting the shape of the EGG waveform itself. Instead, one must take care not to introduce hum in the first place. The placement of the equipment and common electrical grounding of all components will be important. Again, listen to the incoming EGG signal when the electrodes are in place. If a hum changes audibly when you touch your computer or the EGG device, then the electrical grounding needs attention. A first step would be to make sure that all interconnected equipment is supplied from the same power outlet. If the hum persists, disconnect from the mains and power bricks and run all devices on battery power. This is also good for electrical safety. Sometimes a hum will go away if you just rearrange the equipment or relocate your setup to another room. Although the same considerations apply to the microphone signal, EGG electrodes may be less shielded electromagnetically than microphones. 

### 3.5. Low-Pass Filtering

Figure 6 shows an example of a quiescent EGG signal with a small 50 Hz hum at about −70 dBFS (dB relative to full scale). There is also a noise band around 18 kHz that pulsates in synchrony with the hum, so it may be an artifact of a switching power supply. The spectrogram in Figure 6b was made with +6dB/octave pre-emphasis (as for the derivative), and it can be seen that these very high frequencies will dominate the derivative, so they must be eliminated. EGG spectra seldom contain any interesting information at very high frequencies, so a steep low-pass filter at 10 kHz will suffice to do so. Whether or not the low-frequency hum will be a problem will depend on the strength of the AC component of the EGG when phonation is present.

### 3.6. Noise Reduction in Audio Production

The above methods can be readily implemented with signal-processing algorithms in any preferred computer language, including Matlab^®^ or Python. In audio production applications, these methods form the mainstay of noise suppression techniques. For instance, the freeware Audacity (www.audacityteam.org, accessed on 24 March 2024) comes with a ‘Noise reduction’ function that performs spectral thresholding to adjustable degrees and in several frequency sub-bands; it also has notch filters of the kinds mentioned above. It has numerous settings that need to be chosen manually for each case. Large software applications called ‘digital audio workstations’ (DAWs) will accept third-party software plug-ins for special purposes, including noise reduction. An example is the ReaFIR plug-in that comes with the popular DAW Reaper (www.reaper.fm, accessed on 24 March 2024). Such plug-ins are typically ‘trained’ on quiescent segments of the signal. From these segments, a model is created of the shape of the noise floor, which is then used as a spectral threshold for the active segments of the signal. Such tools can be useful for dealing with especially noisy signals such as the EGG, but they require experience and many trials to achieve good results.

## 4. Detecting Vocal Fold Contact

The gain of EGG signals is all but impossible to calibrate because the signal strength varies with the physiology of the informant, and with the vertical larynx position changing relative to the electrodes. Therefore, a simple amplitude threshold is not a good criterion for detecting contact. Rather, we need a criterion that in some way considers only the EGG waveform *shape*.

### 4.1. Using Q_Δ_

VF oscillations without contact along the length of the vocal folds, as in breathy voice, tend to modulate the contact area only near the ends of the vocal folds by a very small amount and in a sinusoidal fashion. As soon as the vocal folds make contact somewhere along their length, the rate of increase in the VF contact area increases abruptly. Conversely, if the EGG waveform is very close to sinusoidal, regardless of amplitude, it is very likely that there is no contact. These cases can be discriminated using the peak of the time derivative of the EGG waveform. The EGG derivative (dEGG) is easily estimated using the sample-to-sample differential *δ*. A sinusoidal waveform has a normalized peak derivative *Q*_Δ_ of 1 (the maximum of the cosine function). The expression for *Q*_Δ_ (derived in [5]) of a sampled signal is
*Q*_Δ _= 2*δ_max_*/(*A_p-p_*·sin(2π/*T*))(1)
where *A_p-p_* is the peak-to-peak amplitude of the EGG, *T* is the period time in sample intervals, and *δ_max_* is the largest sample-to-sample differential observed over a cycle. In practice, there seems to be a reliable contacting criterion at *Q*_Δ_ > 2. 

The differential *δ* is, however, quite susceptible to noise. Any noise or spurious high frequencies will interfere, by causing the peak derivative of noise plus signal to be randomly larger than that of the EGG signal alone. Simply low-pass filtering the signal will not remove noise below the filter cutoff frequency and has the further disadvantage that attenuating high harmonics will also reduce the estimate of *Q*_Δ_. Even with good EGG hardware, the system noise at these low amplitudes will prevent *Q*_Δ_ from descending completely to 1, which obscures the transition between contacting and non-contacting. This is one reason why we are so interested in de-noising the EGG.

### 4.2. Using the HRF

The ‘harmonic richness factor’ (*HRF*) was defined by Childers and Lee [15] as
(2)$$HRF=(\sum_{i>2}^{10}H_{i})/H_{1}$$
where *H_i_* is the amplitude of the *i*th harmonic and *H*_1_ is the amplitude of the first harmonic, at the fundamental frequency. While their interest was in the glottal flow waveform, the *HRF* metric is equally applicable to the EGG waveform, then denoted *HRF_egg_*. An advantage of the *HRF_egg_* over *Q*_Δ_ is that in high falsetto voice, *Q*_Δ_ can become rather low, obscuring the contacting transition, while the *HRF_egg_* remains greater than −10 dB. The main disadvantage of the *HRF_egg_* is that we need to transform the EGG signal into the frequency domain and find the harmonic peaks. 

## 5. A Practical Implementation

In order to visualize how metrics of the acoustic and EGG signals vary over the range of the voice, it is useful to make voice maps [16]. The public-domain software FonaDyn [17] is a voice mapping system in continuing development. FonaDyn is written in SuperCollider [18], which is an open-source system developed for creating computer music. SuperCollider (SC) comprises a language, a signal-processing server, and a development environment, and it comes with an extensive library of high-level functions that facilitate the creation of real-time sound-processing applications. Under the hood, FonaDyn implements all the signal conditioning of the EGG signal mentioned above, as follows. 

### 5.1. High-Pass Filtering 

In FonaDyn this is done using a fixed 1024-point FIR filter, previously designed in Matlab^®^ to have a linear phase response (the same delay at all frequencies), and the magnitude response shown in Figure 2. Its table of filter coefficients is included in the SC source code for FonaDyn. The filter class ‘Convolution2’ uses a fast Fourier transform (FFT) to expedite the filtering. The SC code for this is as follows.


/* Setting up: 1024 FIR coefficients pre-generated in Matlab for highpass @ 100 Hz */
  hpCoeffs = VRPSDIO.getCoeffs(1);
/* Allocate a buffer on the DSP server and pre-fill it */
  hpBuffer = Buffer.sendCollection(Server.default, hpCoeffs, 1, -1);  
/* When running: Get rid of as much near-DC as possible with HP @100 Hz */
  eggCond = Convolution2.ar(inEGG, hpBuffer. bufnum, 0, 1024);

### 5.2. Notch Filtering

This uses zero or more parametric filters of the class BPeakEQ, which uses only one second-order section. After examining the narrow-band spectrum of the quiescent EGG, the user can add such filters, each with a single line of text. The frequency, gain, and Q factor are given in a configuration statement. 


/* Specify zero or more notch filters against stationary tones/bands */
 FonaDyn.config(addEGGNotchFilter: [freq, level, Q]). // add one line for each notch  
/* When starting, FonaDyn instantiates the filters in the list ‘notchFilters’, in series */
 notchFilters.do { | p |  eggCond = BPeakEQ.ar(eggCond, freq: p[0], rq: p[2].reciprocal, db: p[1]);   format("Applying notch filter to EGG: % Hz, % dB, Q=%", p[0], p[1], p[2]).postln; };

### 5.3. Spectral Thresholding

Using the built-in DSP functions, only a few lines of SC code are needed for spectral thresholding. The user sees only a ‘De-noise’ number field for controlling the threshold (Figure 7). ‘*thresh* = 0’ means no thresholding; *thresh* > 0 controls the threshold level.


chain = FFT(LocalBuf(2048, 1), eggCond); // To freq. domain; half-sine window by default

thresh = In.kr(ciBusThreshold);               // Get the threshold value from the GUI

chain = PV_Compander(chain, thresh, 4.0, 1.0); // 4.0 is the dB expansion ratio

eggCond = IFFT(chain);                       // Back to the time domain

micCond = DelayN.ar(micCond, 0.075, 0.075);   // Keep audio and EGG in sync


With an *N*-point FFT, the level of the spectral noise threshold becomes 20 × log_10_(*thresh*/*N*) (dB down relative to full scale). A half-sine windowing function is used for the FFT because it gives a somewhat narrower bandwidth than a Hann or Hamming window. In principle, it could be possible to combine all the filtering operations and the spectral thresholding using a sequence of pre-configured operations on a single frequency-domain buffer, thereby somewhat reducing the computational load and the delays. This optimization remains to be implemented.

### 5.4. Harmonic-Domain Analysis

If the EGG signal can be accurately segmented into periods, a direct Fourier transform can be applied in the time domain over exactly each period. In this way, as many harmonic magnitudes (and phases) as desired can be obtained directly for each cycle. The shape of each EGG pulse can then be reconstructed, practically without noise. In effect, such harmonic decomposition and reconstruction amounts to yet another kind of de-noising. The end result is similar to that from a brick-wall low-pass filter that tracks *f*_o_ and passes exactly *N* harmonics, but this is possible only if a reliable period estimate is already available.

FonaDyn does have quite a robust period segmentation [4], and it characterizes EGG shapes using Fourier decomposition, so the *HRF_egg_* as defined in Equation (2) can be computed cycle-by-cycle without the need to search for spectrum peaks. However, it is too challenging in the SuperCollider formalism to concatenate the reconstructed cycles back into a contiguous signal in real time. Informal testing has shown that the onset of VF contacting appears to correspond to the *HRF_egg_* (in decibels) becoming higher than about −10 dB, but more experimentation is needed. With vocal effort increasing from breathy to loud voice, *HRF_egg_* reaches a maximum of about +5 dB when the contact quotient *Q*_ci_ [4] is at its smallest (with the most overtones), and then decreases again as the EGG pulses become wider.

### 5.5. Results

The primary use case of EGG de-noising will be to remove low-level system noise from an otherwise clean recording, in order to detect whether or not VF contacting is present. Figure 7 and Figure 8 are montages of partial screen-dumps from FonaDyn. The middle panels show the EGG waveform with the amplitude and the cycle time both normalized to [0…1]. The snapshot of the screen was taken at an instant of non-contacting. The left panels show the immediately preceding episode of very soft phonation in which vocal fold contacting comes and goes within 1.6 s. The contact quotient *Q*_ci_ (red trace) is defined as the area under the normalized pulse, i.e., 0.5 for a sine wave, and the normalized peak dEGG *Q*_Δ_ (yellow trace) is the maximum positive slope, which is found at the very edges of the graph. Note how in (b), the yellow curve becomes much clearer and descends to 1. This happens quite abruptly when contacting ceases. The right-hand panels show voice maps of *Q*_Δ_ using a color scale from green for 1 (sine wave) to red for 20 (very rapid contacting). Note in (b) how the bright green region of non-contacting becomes much more distinct. In Figure 7 and Figure 8, there are also histograms of *Q*_Δ_ based on the voice map cell averages.

A rule of thumb is that the de-noising has been effective if histograms of *Q*_Δ_ are clearly bimodal, with one distribution peak near 1, corresponding to breathy, non-contacting phonation, and another peak higher than about 4, corresponding to well-established VF contacting. This is on the condition that the informant was successfully instructed to exercise the voice range across the transition between soft/breathy and normal phonation. Phonations of sustained vowels at only a few effort levels will not be sufficient for testing this. An upcoming companion article will show how the onset of vocal fold contacting is an important complement to the interpretation of several other voice metrics.

**Figure 7 bioengineering-11-00479-f007:**
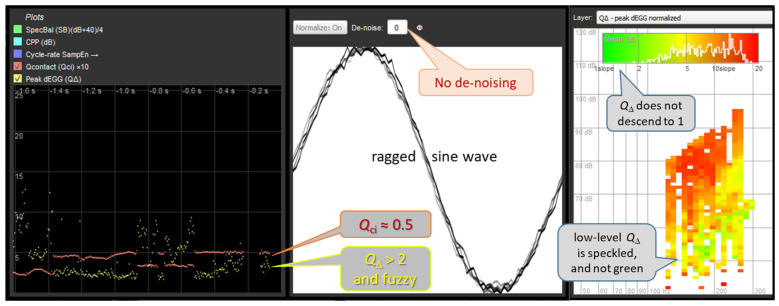
An example of the EGG in very soft phonation. **Left**: time tracks of the *Q*_ci_ (red) and *Q*_Δ_ (yellow). Each fleck represents a value from one phonatory cycle. Note how *Q*_Δ_ is erratic because of low-level noise. **Center**: appearance of a sinusoidal EGG signal in the absence of contacting. **Right**: voice map of *Q*_Δ_ of an amateur male singer performing crescendo–decrescendo on notes of a scale. Each ‘pixel’ in the maps is one semitone wide and one decibel high.

**Figure 8 bioengineering-11-00479-f008:**
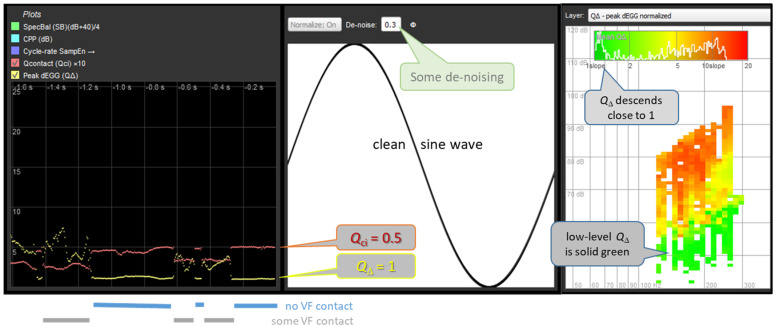
The same example as in Figure 7, with appropriate de-noising applied. **Left**: Note how *Q*_Δ_ of the de-noised signal is lower and less erratic overall, and how it descends properly to 1 when there is no vocal fold contact. **Center**: appearance of a sinusoidal EGG signal in the absence of contacting. **Right**: voice map of *Q*_Δ_ of the same recording, with de-noising. The upper border of the green area corresponds to the onset of vocal fold contacting, which becomes clear only with de-noising. Illustration adapted from the FonaDyn Handbook [18] and used with permission.

To assess the efficacy of the de-noising with regard to the distribution of *Q*_Δ_, a simulation of EGG signals was carried out with Gaussian white noise added and then removed. The outcome is shown in Figure 9. The panels in the right-hand column of Figure 9 show how the de-noised versions approach the noise-free case.

## 6. Difficult Cases

It can happen that side tones are not static but wander in frequency, which makes it harder to suppress them. Wandering tones are usually easy to hear and will also stand out in a spectrogram of the EGG signal (Figure 10). Any EGG hardware with moving interference tones is likely to be poorly engineered and should be avoided. In this example, it is also seen that the device in question has an automatic gain control, which is not helpful at all for spectral thresholding. The threshold would have to adapt to the changing gain. A recording such as this would require more ambitious processing than what has been presented here.

## 7. Discussion

The types of EGG signal conditioning detailed here—steep high-pass filtering, spectral thresholding, and notch filtering—will all improve both the precision of any subsequent cycle segmentation and of the estimate of metrics of the EGG waveform. In particular, computing the contact quotient *Q*_ci_ as described in [4] needs a really DC-free signal, and computing *Q*_Δ_ as in [4] requires a high SNR. Conversely, *Q*_ci_ is not very sensitive to the SNR, and *Q*_Δ_ is insensitive to the near-DC content.

The fewer the harmonics in the signal, the lower the peak derivative will be. *Q*_Δ_ therefore depends on the number of harmonics within the channel bandwidth, and will tend to decrease with increasing *f*_o_. If this dependency is problematic, it can be removed by first constraining the EGG signal to a fixed number of harmonics, whose highest frequency never exceeds the bandwidth of the channel. This issue may require attention for systems that band-limit the signal to 10 kHz or less.

In the spectra given here, the spectrum levels often seem to descend well below the quantization noise floor, which at a 16-bit sample depth would be about −96 dB. The reason is that the quantization noise power in each frequency bin is the total quantization noise power divided by the number of frequency bins, which is typically 2048 = 2^11^. Since the digitized signals are amply represented internally by floating-point numbers with at least 32 bits, this lowers the noise floor in each bin by 11 × 3.01 ≈ 33 dB.

Power line hum is all too common, so it is annoying that getting rid of it is harder than one would think. As a curiosity, we note that, in principle, a stable electrical hum could be analyzed from non-phonated segments and resynthesized in counter-phase to the entire recording so as to remove the hum by cancellation. This would accurately restore the EGG waveform. However, in utility power grids, variations in the total nationwide load will slightly perturb the mains frequency from the nominal 50 or 60 Hz by up to ±0.1 Hz (normal deviations) or ±0.5 Hz (abnormal deviations). The hum frequency would need to be determined very accurately and to remain completely stable over a useful length of time (minutes). Such a cleaning effort would be motivated only for salvaging very important recordings.

## 8. Summary

Frequency-domain thresholding with dynamics expansion retains the strongest frequency components in a signal and suppresses everything else.It improves the dEGG considerably, which in turn clarifies the onset of VF contacting.The EGG waveform essentially retains its shape (unlike with time-domain filtering).It works best against white system noise, with a flat spectrum.A single number suffices to specify the expansion threshold level. In FonaDyn, this number can be adjusted interactively for the best effect while observing and listening to the signal.Non-white noises such as hum and spurious side tones are also suppressed if they are below the threshold. This can often be achieved by applying tailored notch filters before the threshold-expand operation.

## 9. Conclusions

A static frequency-domain threshold and accurate application of optional notch filters are often sufficient to make a noisy EGG signal usable for the detection of the onset of vocal fold contacting. As will be shown in a separate article, such detection is important for interpreting other voice metrics over the voice range.

## Figures and Tables

**Figure 1 bioengineering-11-00479-f001:**
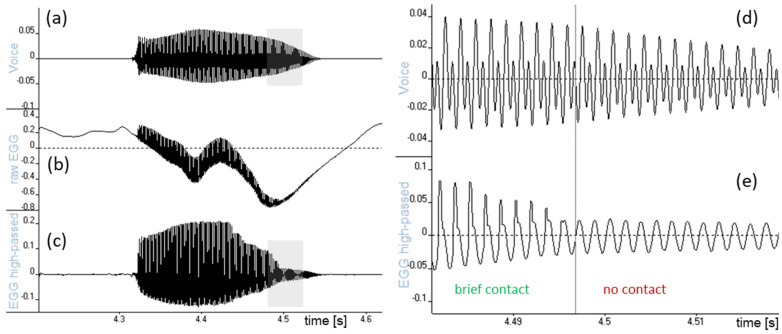
Views of a brief high-pitched syllable /a/. Left: (**a**) the acoustic voice signal; (**b**) the ‘raw’ EGG signal with near-DC fluctuations remaining after a mild analog high-pass filter in the device; (**c**) the scaled-up EGG signal after steep digital high-pass filtering. Right: expanded views of the shaded portions: (**d**) voice and (**e**) high-pass-filtered EGG. Note in (**e**) how the EGG signal quickly becomes sinusoidal when the brief contacting in every cycle ceases, while there is no corresponding abrupt change in the acoustic signal (**d**). The horizontal axis is time in seconds, the vertical scale is amplitude relative to full scale. In the EGG plots, vocal fold contact area increases upwards.

**Figure 2 bioengineering-11-00479-f002:**
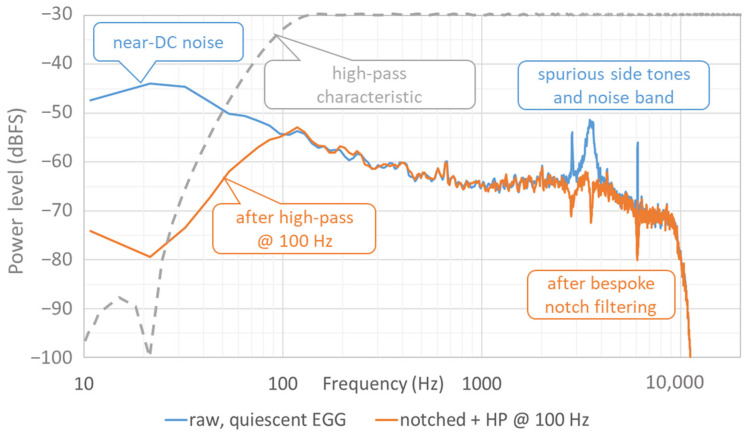
Narrow-band LTAS of an EGG signal during a pause in phonation, i.e., of only background noises. Blue line: original ‘raw’ signal; orange line: after digital high-pass filtering with a 1024-point, high-pass linear-phase FIR filter (dashed grey line) to suppress near-DC content, and after three notch filters, manually added, at 2850, 3520, and 6080 Hz.

**Figure 3 bioengineering-11-00479-f003:**
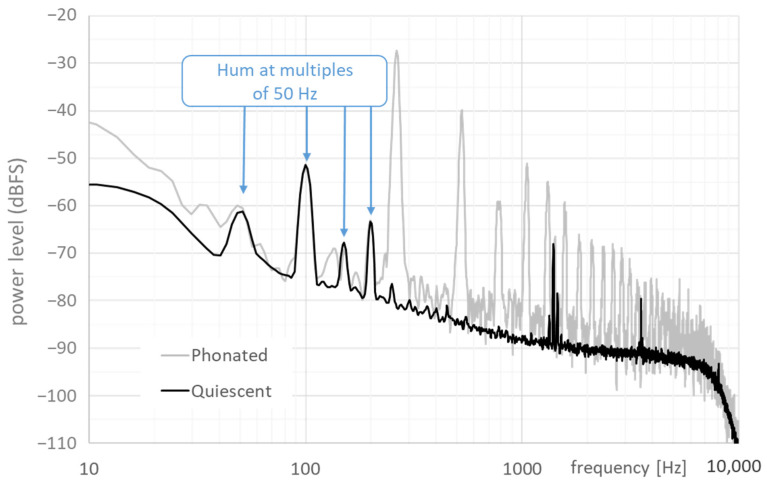
Black line: LTAS of an example quiescent segment (no phonation); gray line: LTAS of a segment with phonation. The harmonics of the hum can be strong enough to distort the waveform of the EGG. Here, we also see stray frequencies at around 1300 and 3500 Hz.

**Figure 4 bioengineering-11-00479-f004:**
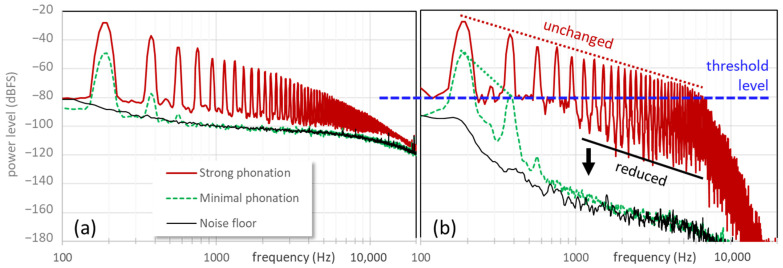
The effect of thresholded dynamics expansion on the spectrum: (**a**) spectra of the EGG signal in strong and minimal phonation, and the system noise floor without phonation; (**b**) a level threshold is applied at −80 dBFS (dB relative to full scale), and levels below the threshold are expanded downwards. Note how harmonics above the threshold are unchanged, essentially preserving the EGG waveform in the time domain, while noise above 1 kHz is considerably attenuated, by 20–35 dB between harmonics.

**Figure 5 bioengineering-11-00479-f005:**
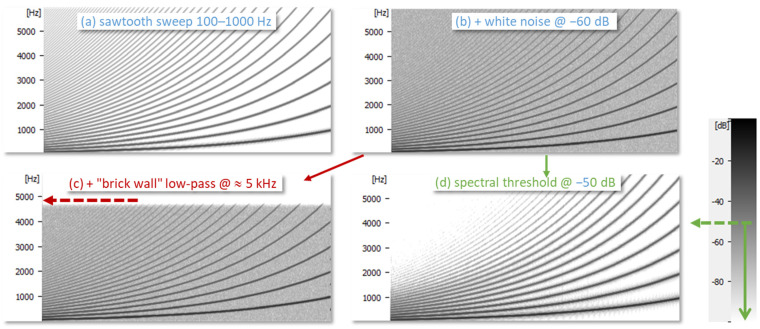
Spectrograms of a swept sawtooth signal (**a**) plus white noise (**b**), to illustrate the difference between low-pass filtering (**c**) and spectral thresholding (**d**).

**Figure 6 bioengineering-11-00479-f006:**
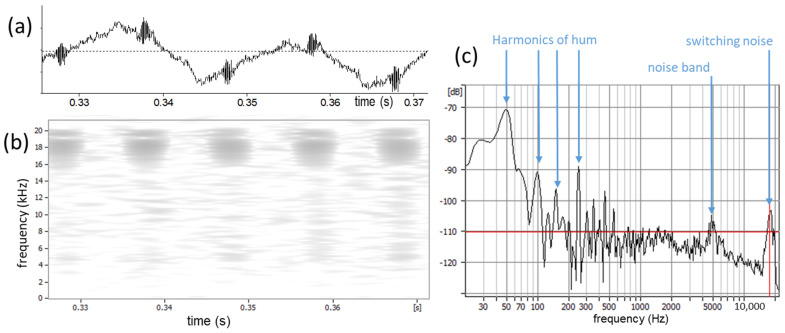
A quiescent portion of an EGG signal, to illustrate noise phenomena. (**a**) Low-level 50 Hz hum at high magnification, with two noise bursts per cycle. (**b**) The corresponding spectrogram, with a +6 dB/octave pre-emphasis, which corresponds to the effect of taking the derivative. Note the band of noise at about 18 kHz, which can be removed with a low-pass filter. (**c**) Spectrum section showing the harmonic components of this hum, the noise peak at 18 kHz, and another around 5 kHz.

**Figure 9 bioengineering-11-00479-f009:**
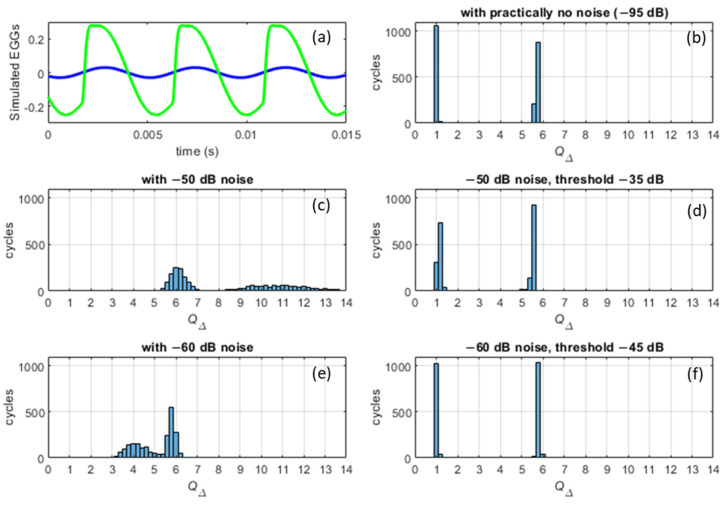
Using simulated EGGs for testing the spectral thresholding with regard to the detection of VF contacting. (**a**) Sinusoidal waveform with *Q*_Δ_ = 1 at −33 dBFS (blue) and with *Q*_Δ_ ≈ 5.8 at −13 dBFS (green). The *f*_o_ is 220 Hz. (**b**–**f**) Histograms of the *Q*_Δ_ of the two waveforms alternating with a one-second duration each, repeated five times. (**b**) The ‘ideal’ no-noise distribution with only the values 1 and 5.8 represented. (A dither noise at −95 dB was, however, applied, to avoid taking the log of zero.) (**c**) A constant Gaussian white noise is added at −50 dBFS. (**d**) Ditto, with a spectral threshold applied at −35 dBFS. (**e**) A constant Gaussian white noise is added at −60 dBFS. (**f**) Ditto, with a spectral threshold applied at −45 dBFS. Note how the *Q*_Δ_ distribution with even a little noise (**e**) is quite wrong, while the de-noised ones (**d**,**f**) are close to the ‘ideal’ in (**b**), thus clearly indicating VF contacting when *Q*_Δ_ ≫ 1.

**Figure 10 bioengineering-11-00479-f010:**
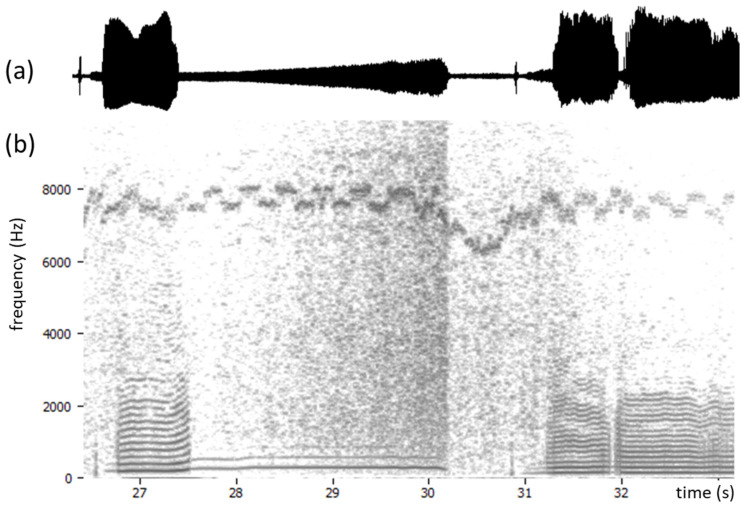
(**a**) EGG signal envelope and (**b**) spectrogram from an EGG device with automatic gain control, as can be seen operating in the interval 28–30 s. This makes the use of a stationary spectral threshold impossible. Also, in the 6–8 kHz region, there is a wandering side tone that cannot be negotiated with a static notch filter.

## Data Availability

No new data were collected for this Technical Note. Examples were pulled from existing recordings, some of which have not yet been analyzed for publication; however, the point is only to demonstrate different kinds of noise in EGG signals, without regard for any informant-specific information.

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
