# Peer review of "Pragmatic De-Noising of Electroglottographic Signals"

_bioengineering, 2024, doi:10.3390/bioengineering11050479_

Round 1

Reviewer 1 Report

Comments and Suggestions for Authors

The submission paper is a technical note that presents the pragmatic de-noising methods for electroglottographic (EGG) signal analysis. The text describes different types of noise artifacts in EGG signals, and signal preprocessing techniques, including the spectral thresholding and static notch filtering.

The following comments could be considered to improve the manuscript:

1) The title of Section 3 could be revised as "Signal preprocessing techniques".

2) Regarding the high-pass, notch, and low-pass filtering methods, the typical design of filter parameters should be provided, such as how to determine the filter order, the frequency cutoff, and bandwidth settings.

3) It is suggested providing an example of "black box" application to demonstrate the effects of a third-party software plug-ins on acoustic signal processing related to EGG analysis.

4) The resolution of Figures 7 and 9 should be improved to at least 300 dpi as printing level. The current figures are vague for display.

5) In addition to the textbook techniques, it is suggested more descriptions on the state-of-the-art technologies on EEG signal preprocessing and parameter extraction, with more recent references in the literature.

Author Response

x

Reviewer 2 Report

Comments and Suggestions for Authors

The article provides an overview of methods for processing EGG signals. However, it does not meet the standards expected of academic publications.

Shortcomings:

1. The article does not show originality, presenting conventional signal processing techniques without reviewing any novel methods or approaches (see Best Conferences - A*).

2. The language used in the article tends to be unnecessarily complex, making it inaccessible to researchers unfamiliar with advanced signal processing concepts. This hinders comprehension and reduces the effectiveness of the communication of ideas.

3. The content lacks depth and fails to provide substantive insights into EGG signal processing. The missing detailed discussions and comparative analyses with existing methods diminish the scientific value of the manuscript.

4. The results presented in the article are not convincing and do not demonstrate significant advances or improvements over existing techniques (see best conferences - A* and Q1 journals).

5. The absence of a comprehensive review deprives researchers of essential background and context necessary to understand the significance of the techniques presented. This omission undermines the credibility and scientific rigor of the article.

6. There is no conclusion.

7. Only 9 references, that's very few. No description of the best papers/articles that are consistently presented at highly ranked conferences and journals Q1.

Comments on the Quality of English Language

Extensive editing of English language required.

Reviewer 3 Report

Comments and Suggestions for Authors

Denoising techniques play a pivotal role in the processing and analysis of biosignals. This technical note introduces a practical denoising strategy tailored for EGG signals. Here are some pertinent comments for consideration:

  1. In Section 1, it would be beneficial to expand on related works that delve into various denoising methodologies, such as wavelet-based techniques, pertinent to biosignals. Including additional references could augment the impact of this note.

Aviles-Espinosa, R.; Dore, H.; Rendon-Morales, E. An Experimental Method for Bio-Signal Denoising Using Unconventional Sensors. Sensors 202323, 3527. https://doi.org/10.3390/s23073527

Boyer, M.; Bouyer, L.; Roy, J.-S.; Campeau-Lecours, A. Reducing Noise, Artifacts and Interference in Single-Channel EMG Signals: A Review. Sensors 202323, 2927. https://doi.org/10.3390/s23062927

Kim, J.-M.; Kim, M.-G.; Pan, S.-B. Study on Noise Reduction and Data Generation for sEMG Spectrogram Based User Recognition. Appl. Sci. 202212, 7276. https://doi.org/10.3390/app12147276

Chien, Y.-R.; Wu, C.-H.; Tsao, H.-W. Automatic Sleep-Arousal Detection with Single-Lead EEG Using Stacking Ensemble Learning. Sensors 202121, 6049. https://doi.org/10.3390/s21186049

Jovic, A. Intelligent Biosignal Analysis Methods. Sensors 202121, 4743. https://doi.org/10.3390/s21144743

Z. Liu, J. Chang, H. Li, L. Zhang and S. Chen, "Signal Denoising Method Combined With Variational Mode Decomposition, Machine Learning Online Optimization and the Interval Thresholding Technique," in IEEE Access, vol. 8, pp. 223482-223494, 2020, doi: 10.1109/ACCESS.2020.3043182.

Chien, Y.-R.; Hsu, K.-C.; Tsao, H.-W. Phonocardiography Signals Compression with Deep Convolutional Autoencoder for Telecare Applications. Appl. Sci. 202010, 5842. https://doi.org/10.3390/app10175842

2. In Section 2.1, it's noted that the symbol f_0 is undefined. Additionally, it would be beneficial to include references for "post-compensation" approaches to ensure the note is self-contained.

3. Regarding Fig. 1, clarification is needed regarding the unit of the x-axis, confirming it represents a time index.

4. In Fig. 3, the note mentioning "hum at multiples of 50 Hz" may not be accurate. It's suggested that the frequencies be verified and potentially added to the figure or caption.

5. It's customary to employ a metric index to evaluate the efficacy of denoising approaches. However, this technical note seems to lack such a description or definition. Consider including a discussion on relevant metrics for assessing denoising effectiveness.

Round 2

Reviewer 1 Report

Comments and Suggestions for Authors

The manuscript has been greatly improved and responsed to my comments and review concerns.

Reviewer 3 Report

Comments and Suggestions for Authors

In this revision, all my previous concerns have been well addressed. No further comments for this revsion.